# Applicable Strains, Processing Techniques and Health Benefits of Fermented Oat Beverages: A Review

**DOI:** 10.3390/foods12081708

**Published:** 2023-04-19

**Authors:** Qian Yu, Jiaqin Qian, Yahui Guo, He Qian, Weirong Yao, Yuliang Cheng

**Affiliations:** 1State Key Laboratory of Food Science and Technology, Jiangnan University, 1800 Lihu Avenue, Wuxi 214122, China; 2Collaborative Innovation Center of Food Safety and Quality Control in Jiangsu Province, Jiangnan University, 1800 Lihu Avenue, Wuxi 214122, China; 3School of Food Science and Technology, Jiangnan University, 1800 Lihu Avenue, Wuxi 214122, China

**Keywords:** fermented oat beverages, strains, pre-treatment processes, bioactive compounds, functionality

## Abstract

Based on the high nutrients of oat and the demand of health-conscious consumers for value-added and functional foods, fermented oat beverages have great market prospects. This review summarizes the applicable strains, processing techniques and health benefits of fermented oat beverages. Firstly, the fermentation characteristics and conditions of the applicable strains are systematically described. Secondly, the advantages of pre-treatment processes such as enzymatic hydrolysis, germination, milling and drying are summarized. Furthermore, fermented oat beverages can increase the nutrient content and reduce the content of anti-nutritional factors, thereby reducing some risk factors related to many diseases such as diabetes, high cholesterol and high blood pressure. This paper discusses the current research status of fermented oat beverages, which has academic significance for researchers interested in the application potential of oat. Future studies on fermenting oat beverages can focus on the development of special compound fermentation agents and the richness of their taste.

## 1. Introduction

As one of the most representative plant-based foods, an oat beverage, rich in nutrients and functionalities, has been proven to be a relatively low-cost and excellent substitute for traditional dairy products, and of great help to alleviate lactose intolerance and allergies probably caused by the latter [1]. It contains a variety of nutrients including protein, fat, phenolic compounds, minerals, which make the difference in the prevention of cardiovascular disease, colon cancer, type 2 diabetes and many other diseases [1,2,3,4]. At the same time, oat is rich in dietary fiber; the main component is β-glucan, which can reduce cholesterol and postprandial blood glucose levels, as shown in Figure 1. Another important role of β-glucan is that under the action of intestinal flora fermentation, it can form short-chain fatty acids that protect colonic mucosa, such as propionic acid, butyrate and acetate. These fatty acids can be used as probiotics to promote the growth of beneficial microbiota, support host health, regulate lipid metabolism disorders and reduce the production of oxygen free radicals [2,5]. To lower cholesterol, it is recommended to take 3 g of β-glucan per day [6].

Fermentation is a traditional food processing method that can improve the nutritional value, taste and aroma of food, and extend the shelf life [11]. Under specific conditions, fermentation can multiply a large number of beneficial microorganisms, reduce the content of harmful microorganisms and convert the original macromolecular substances in food into small molecules, which is more conducive to the digestion and absorption of the human body. Scientific research in recent years has confirmed that some microorganisms in the fermentation process can promote health, such as regulating intestinal flora, inhibiting bacteria and anti-inflammatory, and improving immunity [12,13,14,15]. At present, fermentation has been used in milk, fruit and vegetable juice, plant protein, grain, etc. Oats contain large amounts of indigestible oligosaccharides, peptides and essential amino acids to promote microbial growth, which can be used as a good culture matrix for microbial growth, reproduction and biological activity maintaining [16]. Iron, zinc, magnesium, calcium and protein in oats can combine with phytic acid to form anti-nutritional factors, which are not conducive to human absorption. Microbial fermentation can indirectly reduce phytic acid and increase the content of soluble iron, zinc and calcium ions [17]. At the same time, oat will form a variety of volatile flavoring substances (including acetic acid, butyric acid, etc.) in fermentation, which improves the flavor and taste of the products.

The previous studies on oats were mainly about oat raw materials, oat processing, etc., but rarely involved fermented oat beverages. Therefore, this paper provides a comprehensive review of fermented oat beverages, including the strains that can be used and the preprocessing methods. More importantly, it summarizes the nutrition and function of oat beverages after fermentation. We also attempt to present the current challenges and future research directions, hoping to strengthen scientific research and product development to ultimately meet the rapid growth of market demand for fermented beverages.

## 2. Strains

The development of fermented oat beverages started in Europe with the rise of the functional food market. In Sweden, there are two popular fermented oat beverages, which are named Proviva, fermented with *Lactobacillus plantarum 299v*, and Adavena^®^ M40, fermented with two different yoghurt cultures. In addition, as a non-dairy fermented beverage, Adavena^®^ M40 was comparable to yoghurt regarding consumer acceptability [18,19]. Bioferme of Finland launches YOSA, a beverage made from oat and husks which is prepared by fermentation and mixed with probiotics (*Lactobacillus bifidobacterium BB-12* and *Lactobacillus acidophilus LA-5*) [18]. So far, there are still few products in the fermented oat beverage market, but more and more theoretical studies have gradually started to focus on fermented oats.

Oat contains large amounts of indigestible oligosaccharides, peptides and essential amino acids, which can be used as a good culture matrix for microbial growth, reproduction and biological activity maintaining [2]. Vera Maria Klajn et al. developed a fermented soluble oat extract (SOE) beverage using *Lactobacillus casei* (*L. casei*). It is concluded that the production of novel probiotic fermented dairy beverages can be fermented with the combination of SOE and the probiotic *L. casei*, which has been proven to be beneficial for consumers [20]. Furthermore, β-glucan, the primary dietary fiber component in oat, has been shown to be a probiotic, promoting the growth of beneficial microbiota [3,21,22]. Junying Bai et al. investigated the effects of intestinal microorganisms on oat–glucan metabolism in mouse and human fecal microbiota by in vitro fermentation. This experiment confirmed the fermentability of β-glucan in oat and supported its application in probiotics and synbiotics [23].

In the existing studies on fermented oat beverages, the main fermentation bacteria are LAB (including *Lactobacillus*, *Bifidobacteria*, *Lactobacillus*, *Cheeseba*, *Streptococcus*, *Myxobacterium*, etc.), and the main fermentation fungi are yeasts and Candida, as shown in Table 1. These strains enhance the nutrition, digestibility, and organoleptic properties of fermented oat beverages [24].

### 2.1. Lactic Acid Bacteria

*Lactic acid bacteria* (LAB) are commonly used as a fermenting agent for fermented foods. LAB are known for their ability to produce different aromatic compounds due to lipolysis and protein hydrolysis, which contribute to the pleasant organoleptic properties of fermented products. Additionally, LAB produce lactic acid, the presence of which prevent contamination by pathogens and other harmful microorganisms in the product [42]. With the gradual deepening of the study of microorganisms, LAB have recently been linked to vegetables, meat, grains and so on [43].

*L. plantarum* was isolated from human saliva and can be easily found in plenty of plant fermented foods such sauerkraut, Kimchi and bean paste [37]. As one of the most widely studied species of LAB, *L. plantarum* can enhance the intestinal integrity, metabolic activity of intestinal cells and stimulate immune responses [44]. The genome of *L. plantarum* is one of the largest found in LAB, and its complex genome results in a versatile metabolism, high level of adaptability to various environmental niches, and the capacity to grow on plant material with a high phenolic compound content, which has been demonstrated to be available as a starter culture for plant fermentation [45]. Małgorzata Wronkowska et al. compared the fermentation of oat flour with *L. plantarum* and *L. casei* and found that fermentation conducted with *L. plantarum* for at least 12 h enables obtaining a product with high antioxidant activity and high contents of volatile aroma compounds [11]. Natalia Aparicio-Garcí et al. used two strains of *L. plantarum* and one strain of *L. casei* to ferment 5% oat flour suspension for 24 h in order to investigate their kinetic processes during 24 h of fermentation. The authors judged from the growth curves of both strains that *L. plantarum* was better than *L. casei* in oats for the same fermentation time because *L. casei* was not as good at utilizing oat components as *L. plantarum*. Additionally, the authors’ study confirmed that *Lactobacillus* strains could improve the antioxidant activity of oats, in the same way as Sunderam et al. [46]. The formation of volatile compounds was related to the metabolic activity of the strain, and the largest chemical class detected by the authors in the fermented oat samples was aldehydes, followed by alcohols, ketones and acids. These compounds enabled the oats to obtain a better flavor profile. Based on the data obtained, the authors suggested the application of *L. plantarum* W42 for the fermentation of oat flour, allowing for the authors to obtain products with high antioxidant activity and a high content of volatile aromatic compounds [47]. Nionelli Luana et al. fermented oatmeal with *L. plantarum*, *L. casei*. and *Lactobacillus parasitum* (*L. parasitum*), respectively, in which the beverage fermented with *L. plantarum LP09* had the best value and the most balanced sensory properties [40].

Besides *L. plantarum*, *Bifidobacterium lactis* (*B. lactis*), *Lactobacillus rhamnosus* (*L. rhamnosus*), and *Streptococcus thermophilus* have also been demonstrated to be well adapted in oat substrates. *B. lactis* belongs to Bifidobacterium, which is a facultative anaerobic microorganism that naturally exists in the mouth and intestines of mammals and humans. Azadeh Asadzadeh et al. fermented a non-dairy probiotic beverage based on oat bran extract with *B. lactis* (Bb-12) and studied the effects of storing oat bran extract and carbon dioxide at different concentrations for 21 days on the activity of Bb-12 probiotics in beverages. The results showed that bacterial viability increased significantly in the first 14 days with an increase in carbon dioxide concentration, oat bran extract concentration and storage time [33]. *Lactobacillus rhamnosus* (*L. rhamnosus* LGG) mostly exists in the intestines of humans and animals [38]. Bacterial taxonomy belongs to *Lactobacillus*, which cannot utilize lactose but can ferment a variety of monosaccharides (glucose, arabinose, maltose, etc.). *L. rhamnosus* LGG has outstanding performance in stomach acid and bile, can enter the human intestine easily, and has high application value and development prospects in fermented dairy products [38]. Carmen Masiáet al. combined *L. rhamnosus* LGG^®^ with different LAB and Bifidobacterium bifidum BB-12^®^ to ferment oat. By evaluating the acidification, titratable acidity and activity of *L. rhamnosus* and Bifidobacterium, as well as the volatile organic compounds in fermentation samples, it was concluded that LGG^®^ and BB-12^®^ could grow and survive in an oat substrate, and the main effect of BB-12^®^ was observed in fermented oat samples. LGG^®^ increases the perception of acid-related flavor properties and reduces the sweetness and the bad flavor of grains in oat [26]. The texture and flavor of fermented oat beverages dominate consumer acceptance and sustainability. Streptococcus thermophilus is derived from dairy products and considered a “Generally Recognized as Safe“ ingredient because of its ability to increase the viscosity of dairy products, Streptococcus thermophilus is often paired with *Lactobacillus bulgaricus* (*L. bulgaricus*) in some important fermented dairy products, including yogurt and cheese, resulting in its unique taste and stickiness [48]. Previous studies have shown that *Streptococcus thermophilus* TKM3 KKP 2030P is an efficient initial strain for the fermentation of plant-based raw materials. It was used by Anna Goncerzewicz et al. to ferment an oat–banana matrix with the addition of β-glucan to obtain a beverage with more L-lactic acid than D-lactic acid, ensuring that the fermented beverage has good sensory characteristics and a large number of LAB within 4 weeks of cold storage [24].

Other LAB such as *L. fermentum*, *L. paracasei*, *L. acidophilus*, etc. have also been shown to be used in fermented oat beverages. Zhishu He et al. fermented oat substrate with *L. plantarum*, *L. acidophilus*, *L. casei*, *Lactobacillus thermophilus* (*L. thermophilus*) and *Lactobacillus bulgaricus* (*L. bulgaricus*). During the first 12 h of fermentation, the growth activity of the five strains showed significant differences due to the utilization of carbon and nitrogen sources and the adaptability to the substrate [32]. The results showed that *L. acidophilus* and *L. plantarum* in oat substrate were more capable of growing in the oat substrate than the other strains [36]. Liwei Chen et al. used oat flour and honey as the fermentation substrate and fermented *L. fermentum* PC1 for 72 h at 37 °C, improving the total antioxidant capacity and phenolic acid concentration. It indicated that the fermentation of oat powder by *L. fermentum* PC1 was a potential probiotic food [35].

### 2.2. Yeasts

Yeasts, mainly used in baking and brewing (wine, beer, bread fermentation), dominate the microbial composition of many alcoholic food fermentations [49]. In general, LABs utilize various sugars as carbon sources while yeasts can utilize both sugars and lipids [50]. Saccharomyces cerevisiae, especially *C. lipolytica* and *Y. lipolytica*, is an effective producer of lipases and proteases [30,34], allowing for proteins and lipids to be hydrolyzed more quickly to amino acids and fatty acids. Grzegorz Dąbrowski et al. produce 16 new variants of almond- and oat-fermented beverages with 3 strains of LAB and 5 strains of yeasts. The study analyzed the apparent viscosity, volatile compounds and fatty acids composition, indicating that after 48 h of fermentation, acidity increased in both types of beverages. The highest content of minor fatty acids was determined in oat beverages fermented by *L. plantarum PK 1.1* and *Kluyveromyces marxianus* KF 0001. Among the used strains, *Yarrowia lipolytica* YLP 0001 was found to be a major producer of aromas and lactic acid in these two beverages, suggesting that the strain can utilize various carbon substrates [50]. Lavinia Florina Călinoiu et al. evaluated the potential of *Saccharomyces cerevisiae* (*S. cerevisiae*) fermentation in increasing the phenolic acid content and composition, and the antioxidant activity of commercial wheat bran (WB) and oat bran (OB). The results revealed that *S. cerevisiae* was effective for both WB and OB to increase their antioxidant activity by enriching matrices in phenolic acids [31].

However, there are few studies on the application of yeasts in fermented oat beverages. It is possible that yeasts are not as capable of producing acid as LAB, or that the primary byproduct of yeasts consuming sugar is alcohol.

### 2.3. Mixed Strains

In the process of mixed fermentation, yeasts provide monosaccharides, B vitamins, and enzymes for LAB. Lipase and esterase contribute to flavor fermentation in beverages, making them more attractive, while phytase reduces phytic acid, an antinutrient that seriously affects the digestibility of proteins and minerals [51]. Angel Angelov et al. used *L. plantarum B28*, *L casei spp paracasei B29*, *Candida rugosa Y28* and *C lambica Y30* as fermentation strains of oat mud. The cell counts, pH, TA, dry matter, and β-glucan content of fermented oat beverages were determined by the single fermentation of LAB and mixed fermentation of yeasts and LAB. The results showed that the two fermentation methods could reach the appropriate acidity within 6–10 h, and the number of living cells obtained was higher than the level of 10^6^–10^7^ cfu/mL needed for probiotic products, which proved that LAB provided better flavor and shelf life for oat beverages [52]. However, the relationship between LAB and yeasts is complex, and both inhibition and promotion are between them. At present, the symbiosis mechanism of yeasts and LAB at home and abroad is not very thorough.

In the research of fermented oat beverages, we know that LAB are the most ideal strain, improving the original bad flavor of oats to a great extent and making it gradually accepted by consumers. Although many strains of *Lactobacillus* are used for oat fermentation, the strains that can utilize oat ingredients and grow with high viability are mainly *L. plantarum*, *L. rhamnosus* and *L. acidophilus*. Differences in the fermentation characteristics of LAB (such as extracellular polysaccharide production, acid production and flavor production) can change the physicochemical properties and nutritional composition of fermented oat beverages. However, the research on the available strains of fermented oat beverages is not deep enough, mainly focusing on LAB, while the application of yeasts and other bacteria lacks extensive research; therefore, it is necessary to broaden the types of fermentation strains in the future and focus on the development of special strains and compound fermentation agents for fermented oat beverages.

## 3. Processing Techniques

Until now, the process technology studies on fermented oat beverages have mainly concentrated on pre-treatment methods and fermentation conditions. Based on the available studies, the pre-treatment methods of fermented oat beverages (as shown in Figure 2) and the conditions in the fermentation process have been summarized.

### 3.1. Pre-Treatments

Oat, as one of the grains which is rich in soluble fiber, is currently being studied for the ability to improve gastrointestinal function and prevent obesity [53]. Different pre-treatment methods and conditions can obtain different fermentable sugar, which also lead to different bioactive substances in oats and the number and composition of enzymes in the process of enzymatic hydrolysis. There have been many studies on the pre-treatment methods of fermented oat beverages, as shown in Table 2, such as exogenous enzymatic hydrolysis, germination, grinding, malting, and so on.

Oat contains starch, protein, lipids, dietary fiber, minerals and vitamins, among which the starch content is the highest. The shape, size and composition of starch granules can affect the solubility and swelling power of starch, thus affecting the stability of an oat-fermented beverage. Thus, adding amylase and protease to hydrolyze the starch and protein in oats can convert some of them into easily absorbed nutrients like glucose, maltose and amino acids [54]. Enzymatic hydrolysis is a specific and effective method to release bound phenolic acids from cell wall materials, including oats [55]. Cell-wall-hydrolyzing enzymes, such as cellulases, amylase, hemicellulases and pectinase, have been effectively used to release bound phenolics [56]. Qi Bei et al. used cellulase for the continuous hydrolysis of oats with the aim of assessing the potential of cellulase pre-treatment to release insoluble phenolics. The results showed that cellulase could hydrolyze the bonds between phenolics and oat cell wall structures and release insoluble phenolics into soluble phenolics, thus improving the antioxidant activity of oats [55]. Therefore, using enzymatic oat as a fermentation substrate can improve the nutrients of oat beverages, thus has positive implications for the development of fermented oat beverages.

In addition to enzymatic hydrolysis, sprouting is also one of the process methods for oat. Recent studies have shown that germination is an effective way to increase the soluble phenolic content of grains [41,57]. Sprouting is an economical process that can improve the nutritional quality of cereals and their content of bioactive compounds [57,58,59], and can produce more bioavailable vitamins, minerals, amino acids, proteins and phytochemicals than unsprouted grains [60]. Natalia Aparicio-García et al. developed a novel gluten-free fermented beverage based on sprouted oat flour. The results showed that the sprouted oat-fermented beverage (SOFB) is rich in protein (1.7 g/100 mL), β-glucan (79 mg/100 mL), thiamine (676 μg/100 mL), riboflavin (28.1 μg/100 mL) and phenolic compounds (61.4 mg GAE/100 mL), and has high antioxidant capacity (164.3 mgTE/100 mL), but the β-glucan content of SOFB is lower than that of the oat-based fermented beverage containing 7% unmalted oat flour, as reported by Angelov et al. [61], probably because oat β-glucanase hydrolyzes β-glucan during germination and reduces its content [41].

The production or activation of enzymes (e.g., amylase, protease, β-glucanase, etc.), changes in the physical structure of grains and the formation of characteristic compounds related to aroma, flavor and color are the major objectives of malting. Alan Gasiński et al. malted grains of four naked oat varieties (Amant, Maczo, Polar and Siwek) and one covered oat variety (Kozak) in a laboratory setting following typical specifications for barley malt production. The results revealed that malting increased the concentration of phenolic compounds in oat and improved the antioxidant potential of oat [21,62].

**Table 2 foods-12-01708-t002:** Effects of pre-treatments on the composition of oat.

Pre-Treatments	Changes in the Bioactive Components in Oat	References
Enzymatic hydrolysis	Increase soluble and insoluble phenolics. Fermentation following enzymatic hydrolysis upgraded the antioxidant activity and α-glucosidase, α-amylase inhibition activities of oat.	[63]
Sprouting	Obtain a novel gluten-free and healthy ingredient, excellent levels of protein, micronutrients and β-glucan. Present balanced amino acid and fatty acid compositions. Increased free phenolics, GABA and antioxidant activity in oat powder. Present high protease/α-amylase and low lipase activities.	[64]
Malting	Increase the concentration in phenolic compounds. Increase number of compounds present in the malt. Increase the antioxidative potential of grains.	[62]
De-branning	Increase the digestibility rates and decrease phytic acid content significantly. The mineral digestibility raised for all grain samples. Lead to an increase in the protein content of oat flour after separated the bran.	[65]
Drying	Induce starch gelatinization, protein modification. Increase nutrient availability and provide inactivation of heat-labile toxic compounds and other enzyme inhibitors. Lead to a unique sensory and nutritional profile to the oat.	[66]
Milling	Decrease the level of phytic acid.	[67]
	Increase protein digestibility in all flour samples.	[68]
	The oat-milling process is different compared to the milling of other cereal grains as it includes a heat treatment step, namely kilning, which is performed to inactivate the endogenous lipid-degrading enzymes.	[69]
Grinding	Improve the dispersibility, solubility, water retention, antioxidant property, and other important physical and chemical properties	[70]

### 3.2. Fermentation Conditions

It was found that the final state of the fermented oat beverages was influenced by the following factors: strain, temperature, medium concentration, initial pH and oxygen supply. Therefore, the choice of fermentation conditions is particularly important in the production of fermented oat beverages and is also an essential aspect of the final consumer sensory acceptance of the product. Currently, the oat raw materials used to prepare oat-fermented beverages include oat grain, oat flakes and oat flour. Different types of raw materials are used in different water ratios, which affect the viscosity and solid content of oat beverages, thus determining the quality of fermented oat beverages. Natalia Aparicio-Garca et al. determined the soluble protein, β-glucan, and phenol content of a sprouted oat beverage by homogenizing different amounts of sprouted oat flour (5–20% *w*/*v*) and water. It was observed that the content of these compounds increased with increasing amounts of sprouted oat flour. At levels above 18% (*w*/*v*), there was no significant increase in soluble protein–glucan and phenol and the flavor of the beverage was disrupted. Therefore, 18% (*w*/*v*) sprouted oat flour was selected as the best base quality for making beverages [41]. Sumangala Gokavi et al. investigated the fermentation ability of *L. plantarum B-28*, *L. paracasei SSP* and *L. acidophilus* from Kohansen isolated from Bulgarian traditional grain-fermented beverages. It was found that the oat beverage obtained by fermenting an oat substrate at 37 °C for 12 h had better sensory quality than similar products on the market, and this symbiotic fermented beverage based on oats is a functional beverage that provides both probiotics and prebiotics [34].

Fermentation time and temperature are also important factors that must be considered in the microbial fermentation of oat beverages. They determine the taste, color and stability of the beverage, and have an impact on the free phenol and reducing sugar content in it. Fermentation time is very important for enzymes. Generally speaking, enzyme activity and content reach their maximum at the optimum fermentation time, after which they decrease [71]. At the same time, the length of fermentation will affect the acidity of the oat beverages, and excessive fermentation will affect the taste of the oat beverages. Małgorzata Wronkowska et al. fermented oat flour with *L. plantarum IB*, *L. plantarum W42* and *L. casei LCY* for 24 h. It was found that the number of living bacteria was highest in the first 16 to 18 h of oat fermentation, which was positively associated with the acidifying activities of the sample [11]. Fermentation temperature affects the growth of strains to a certain extent and ultimately affects the flavor of the product. The choice of fermentation temperature depends mainly on the characteristics of the fermentation substrate and the fermenting strains. Gupta et al. fermented oat beverage with *L. plantarum* M-13, selecting a fermentation temperature of 37 °C as the optimum growth temperature for *L. plantarum* [16]. Azadeh Asadzadeh et al. used *Bifidobacterium lactobacillus* to produce a functional probiotic beverage from oat bran at a fermentation temperature of 40 °C [33], while the optimum growth temperature for Bifidobacterium is 37–41 °C, and the most suitable temperature for fermentation is 35–40 °C. Therefore, both the optimal growth temperature and the optimal fermentation temperature should be considered when selecting the fermentation temperature.

As seen so far, previous studies on the process of fermented oat beverages have focused on pre-treatment methods, fermentation agents and changes in physicochemical parameters. However, further studies on the optimization of processing conditions for fermented oat beverages are urgently required.

## 4. Health Benefits

### 4.1. Bioactive Compounds

Oat contains a high concentration of bioactive compounds, particularly natural antioxidants such as phenolic compounds [72] and β-glucan [73]. Fermentation has been proven to increase the contents of phenols, anthranilamines and flavonoids in oat and to enhance the protective effect of antioxidants against DNA damage in oat. In addition, fermentation makes antioxidants and micronutrients more readily available and reduces the levels of antinutrients (e.g., phytic acid) and hexanal, which can lead to undesirable flavors [74,75].

Phenolic compounds are produced by several biosynthetic precursors such as pyruvate, acetic acid, some amino acids, malonyl-CoA and acetyl-CoA, through the metabolic pathways of pentose phosphate, mangiferic acid and phenyl propane [76]. Oats contain a variety of polyphenols, such as anthranilamide, hydroxybenzoic acid and hydroxycinnamic acids (i.e., ferulic acid, coumaric acid, coumaric acid and caffeic acid) [72]. Serena Bocchi et al. evaluated the distribution of metabolites in fermented oat beverages digested in vitro. Semi-quantitative analysis showed that fermentation increased the content of saponins and phenolic acids in oats, while phenolic acids increased significantly after fermentation, indicating that fermentation increases the bioavailability of phenolic compounds. One of the reasons for the higher content of 4-hydroxybenzaldehyde and ferulic acid in fermented oats than in unfermented oats is that some of the ferulic acid are bound to the plant cell wall [77], while a class of esterases in LAB are able to release ferulic and other cinnamic acids [78]. The same finding can be found in another work which focused on oat flour beverages, showing that fermentation increases the content of flavonoids and anthraquinones, as well as their antioxidant capacity [57].

Compared with dairy products, the nutritional and sensory properties of oat beverages are sometimes poor, probably owing to the low protein content, the presence of anti-nutritional factors and the lack of some essential amino acids [79]. In order to improve this situation, several processing methods, such as milling, sprouting and fermentation, have been implemented, of which the fermentation process remains the most effective way to improve their sensory, nutritional, and edible properties [80], as shown in Figure 3. Phytic acid is an anti-nutritional factor that inhibits mineral absorption in the small intestine by forming insoluble phytate complexes with charged divalent cations (such as Ca^2+^, Zn^2+^, Fe^2+^ and Mg^2+^) and amino derivatives of proteins at intestinal pH levels. Some LAB have the ability to produce phytase (a phytate-degrading enzyme) during fermentation, which provides optimal pH conditions for the degradation of phytate, leading to phytate dephosphorylation and the increased bioavailability of proteins and minerals [70,71,72,80]. Serena Bocchi et al. [68] demonstrated that oat beverages fermented by *Lactobacillus* spp. and *Bifidobacterium* spp. reduced phytate levels and may lead to the increased bioavailability of micronutrients. Shengbao Cai et al. investigated the activity changes in γ-aminobutyric acid (GABA), phytate and natural antioxidants during the fermentation of oat. GABA, total phenol content (TPC) and flavonoid content increased rapidly, while phytate, as an anti-nutritional component, decreased significantly in fermented oat during fermentation [28,81].

### 4.2. Functionality

Oat has been transformed from an unassuming feed crop to a leading player in the health and functional food ingredient industry, being hailed as the “third staple” [81] because of its nutritional value and therapeutic efficacy on a number of diseases including dyslipidemia, hypertension, vascular damage and diabetes [2,82]. The effects of oats on diabetes are attributed to the properties of maintaining the homeostasis of glucose and insulin levels as well as cholesterol-lowering properties. A variety of fermented foods have been shown to have antidiabetic effects both in vitro and in vivo. Raya Algonaiman et al. used a streptozotocin-induced diabetic rat model to study the potential antidiabetic effects of oat extracts fermented with *L. plantarum* strains; the results of this study showed that the consumption of fermented oats promoted significant antidiabetic and hypolipidemic effects. One of the most important acids significantly released due to fermentation is GABA, a four-carbon non-protein amino acid widely distributed in animals, plants and microorganisms. In addition, the promotion of antidiabetic activity has been mainly attributed to insulin secretion stimulation from pancreatic cells. In the authors’ study, the production of GABA by *L. plantarum* increased significantly in a time-dependent manner. This increase could explain the beneficial effect of fermented oat extracts in attenuating blood glucose levels in vivo [83].

The increased risk of ischemic heart disease is correlated with elevated serum cholesterol levels [84]. Studies show that foods with soluble dietary fiber lower blood lipids by reducing the intestinal absorption of cholesterol and bile acids [41]. Oat contains high levels of soluble fiber, particularly β-glucan [85]; so far, there have been about 50 studies on the effects of oat products on serum cholesterol. These studies have indicated that oat can reduce postprandial lipid levels [16,50,82]. Olof Mårtensson et al. evaluated the effect of fermented oat products on lowering blood lipids. They set up three parallel groups with moderately elevated cholesterol levels, a control group taking fermented dairy products, and two other groups taking fermented oat products and fermented paste oat products. The volunteers who consumed the fermented mushy oat product had a 6% reduction in total cholesterol (*p* = 0.022). This study suggests that fermented mucilaginous oat products containing both natural and microbial glucan can lower blood cholesterol levels [86].

Hippocrates, the Greek physician who is regarded as the father of modern medicine, said, “Let food be your medicine and medicine be not your food.” [87,88]. For thousands of years, people have regarded fermentation as the oldest method for preserving food. At the same time, the study of fermented oat beverages confirmed that fermentation increases the nutritional compounds of oat, lowers the anti-nutritional effects and helps to provide the final beverages with certain antidiabetic and cholesterol-lowering functions.

## 5. Conclusions and Prospects

The emergence of oat beverages as a non-dairy alternative with good taste and unique health benefits offers a new option for people who suffer from lactose intolerance and casein allergies. Bioactive compounds (including peptides and phenolic compounds) were produced. Anti-nutritional factors, such as phytates and trypsin inhibitors, and the hexanal content, which causes undesirable flavors, were decreased during oat beverage fermentation, largely improving the organoleptic properties and nutrition of oat beverages. Currently, the study of fermented oat beverages is still in the initial stages, but its great potential could attract the attention of researchers and the beverage market. However, fermented oat beverages still have problems such as single varieties, poor flavor and an unstable shelf life. Therefore, the future research of fermented oat beverages will focus on: (1) developing the most applicable fermenting agents, (2) optimizing the process, and (3) exploring the functional mechanism of related bioactive compounds. All the findings above will enhance the sensory properties, nutrition and shelf-life of fermented oat beverages and provide a new direction for the development of plant-based beverages.

## Figures and Tables

**Figure 1 foods-12-01708-f001:**
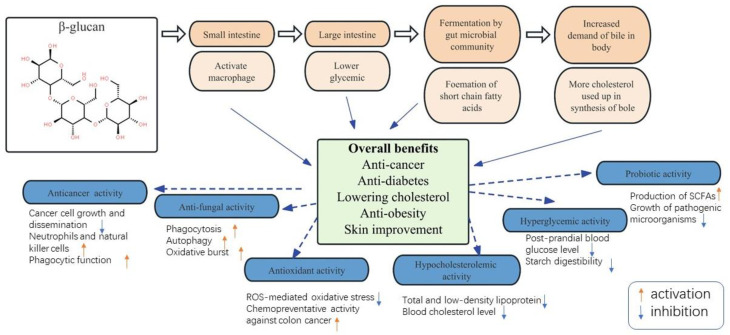
Human health benefits of β-glucan associated with anti-cancer/anti-infection, anti-fungal, antioxidant, hypocholesterolemic, hyperglycemic and probiotic activity [7,8,9,10].

**Figure 2 foods-12-01708-f002:**
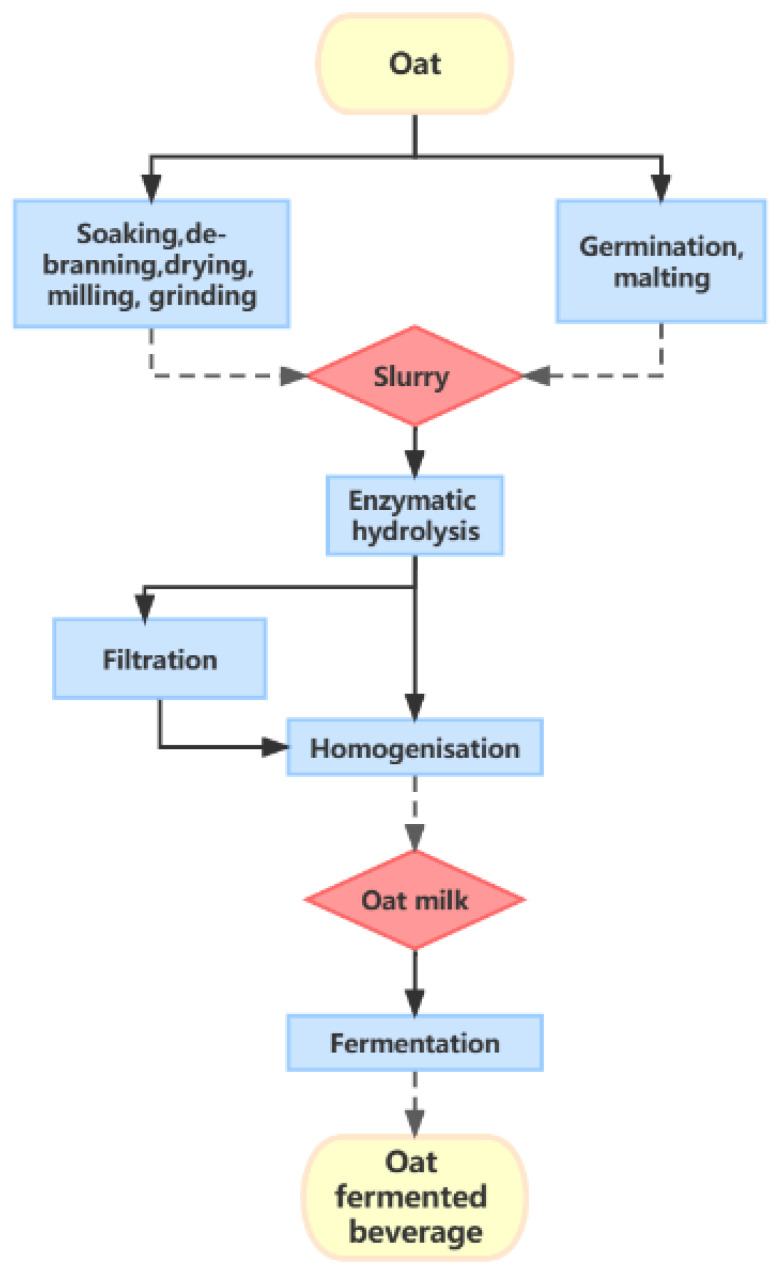
Preparation process of fermented oat beverages.

**Figure 3 foods-12-01708-f003:**
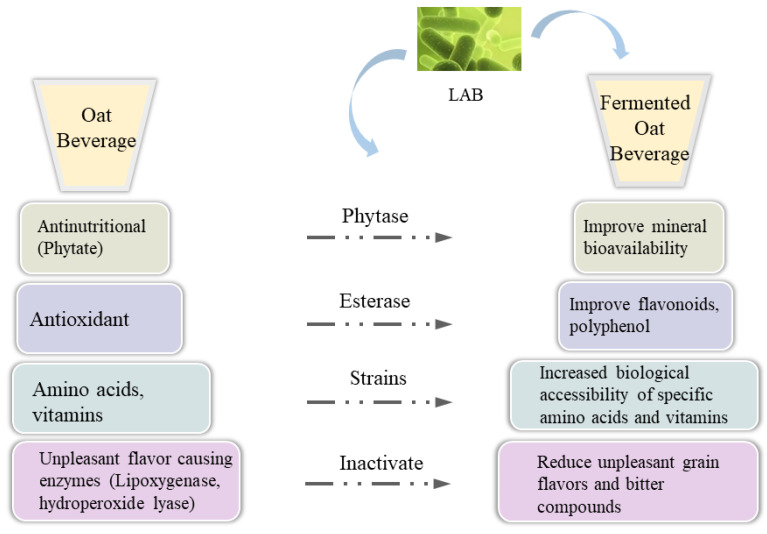
Fermentation improves the nutritional and sensory profile of oat beverages.

**Table 1 foods-12-01708-t001:** Progress in the study of applicable strains of fermented oats and their potential functions.

Strains	Fermentation Substrates	Descriptions	Functions	References
*Lactobacillus rhamnosus GG*	Boil oats mixed with water	Highest growth rate in the oat product. It is able to survive in fermented products during storage at refrigerating temperature, with its metabolic activity continuing.	Anti-inflammatory	[25,26]
*Streptococcus thermophilus TKM3 KKP 2030p*	An oat–banana matrix with Prom Oat additive	Acceptable sensory characteristics and large amounts of lactic acid bacteria were obtained during 4 weeks of cold storage.	Accelerated substrate fermentation	[24]
*Lactobacillus plantarum B28*	Oat mash	Most intensive acid formation was registered for strain *Lactobacillus plantarum B28*, with the pH of the oat medium reaching values below 4.5 after 6 h.	Cholesterol lowering	[27,28]
*L. casei* spp. *paracasei B29*		A good balance between acid formation and cell counts was observed for strains *L. plantarum B28* and *L. casei spp paracasei B29*.	Cholesterol lowering	
*Candida rugosa Y28*		At the end of the fermentation, the highest cell counts were obtained for strains *L. plantarum B28 and C. rugosa Y28* as 2.81 and 2.71 log orders, respectively.	——	[27]
*C. lambica Y30*		*Strain Y30* completed fermentation for 10 h.	——	
*Lactobacillus reuteri* (*NCIMB 11951*)	Oat bran and whole flour	Grow well in the 1–3% pearling fraction, whole flour and bran.	Immunomodulation	[29,30]
*Lactobacillus plantarum* (*NCIMB 8826*)			Immunomodulation Anti-inflammatory	[29,31,32]
*Lactobacillus acidophilus* (*NCIMB 8821*)		The growth limitations of this strain in cereal media.	——	[29]
*Lactobacillus rhamnosus LGG*	Oat concentrate	Acetoin levels increased and acetaldehyde content decreased. No significant effect on rheological behavior was observed when *L. rhamnosus* was present in fermented samples. *L. rhamnosus* significantly enhanced fermented flavor notes, such as sourness, lemon, and fruity taste. Reduce fermentation time.	Improved texture and flavor	[26]
*Bifidobacterium lactis*	Oat bran extract	The viability of probiotic was 10^9^ CFU/mL at pH = 4.2, while by decreasing the pH to 4.0, the viability decreased to 10^7^ CFU/mL.	Anti-diabetes	[33,34]
*Lactobacillus fermentum PC1*	Oats with added honey	Good survival. An increase of more than 50% of gallic acid, catechin, vanillic acid, caffeic acid, p-coumaric acid, and ferulic acid was observed in the methanol extracts. No significant decrease in the β-glucan content was noted during fermentation and storage.	Anti-inflammatory	[35,36]
*Lactobacillus casei* (*431*)	Germinated and malted oat substrates	In the germinated oat media, *Lactobacillus casei* presented the highest maximal growth in this medium. *Lactobacillus casei* showed high adaptability during fermentation. *Lactobacillus casei 431* can produce more typical flavor compounds, such as 2-butanedione, 2-heptanone, acetylurea and 2-nonone.	Improved texture and flavor	[21,37]
*Lactobacillus acidophilus* (*LA-5*)		The oat substrates can support the growth of *Lact. acidophilus*, *Lact. casei* and *Lact. rhamnosus* at probiotic levels comparable to the conventional dairy-based substrates.	——	[21]
*Lactobacillus rhamnosus HN001*			Regulation of intestinal flora	[21,38]
*Lactobacillus rhamnosus IMC 501^®^*	Oat bran	Our study has demonstrated the prebiotic potential of oat bran for *lactobacillus*.	Accelerated substrate fermentation	[39]
*Lactobacillus paracasei IMC 502^®^*				
*Lactobacillus plantarum LP09*	Oat flakes	The beverage started with *L. plantarum LP09*; it had optimal values for all sensory attributes and the most balanced profile.	Improved texture and flavor	[40]
*Lactobacillus plantarum M-13*	Oat flour	It showed excellent survival rate under simulated gastrointestinal tract (GIT) conditions and had a variety of desirable functional properties, such as adhesion, auto aggregation and coaggregation potential, extracellular enzyme production, antibacterial activity and antibiotic sensitivity.	Antibacterial activity and antibiotic sensitivity	[16]
*Lactobacillus plantarum WCFS1*	Sprouted oat flour	The results showed that sprouted oat flour is a suitable substrate that supports the fast growth and high viability of *L. plantarum WCFS1* strain.	Anti-inflammatory	[[35]，[41]]

## Data Availability

No new data were created or analyzed in this study. Data sharing is not applicable to this article.

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
