# Peer review of "Applicable Strains, Processing Techniques and Health Benefits of Fermented Oat Beverages: A Review"

_foods, 2023, doi:10.3390/foods12081708_

Round 1
Reviewer 1 Report
The study provides a comprehensive review of the potential of fermented oat beverages as a value-added and functional food product. The abstract clearly outlines the key points covered in the review, including the applicable strains, processing techniques, and health benefits of fermented oat beverages. The conclusion is well-organized and provides a good summary of the key findings of the study. The authors appropriately acknowledge the current limitations of fermented oat beverages, such as single varieties, poor flavor, and unstable shelf life, and suggest certain areas for future research.
However, it would be useful to provide more specific examples or details regarding the advantages and disadvantages of the processing techniques and strains discussed in the review. Additionally, while the review provides a good overview of the health benefits of fermented oat beverages, it would be beneficial to provide more detail on the specific mechanisms by which these benefits are achieved.
Overall, the study is a valuable contribution to the field and provides useful information for researchers interested in the application potential of oat. The authors should consider incorporating more specific examples and details to further enhance the clarity and usefulness of the review.
Author Response
Point 1:
The study provides a comprehensive review of the potential of fermented oat beverages as a value-added and functional food product. The abstract clearly outlines the key points covered in the review, including the applicable strains, processing techniques, and health benefits of fermented oat beverages. The conclusion is well-organized and provides a good summary of the key findings of the study. The authors appropriately acknowledge the current limitations of fermented oat beverages, such as single varieties, poor flavor, and unstable shelf life, and suggest certain areas for future research.
However, it would be useful to provide more specific examples or details regarding the advantages and disadvantages of the processing techniques and strains discussed in the review.
Response 1:
We appreciate the interest expressed by you in our review as well as the concern raised. More specific examples and details regarding the advantages/disadvantages of the processing techniques and strains discussed in the review have been added according to your suggestion (lines 102-132,216-227,244-256).
Point 2:
Additionally, while the review provides a good overview of the health benefits of fermented oat beverages, it would be beneficial to provide more detail on the specific mechanisms by which these benefits are achieved.
Response 2:
Thanks a lot for your respective comment. More information related to the mechanisms by which the health benefits of fermented oat beverages are achieved have been added (lines355-367,376-389).
Point 3:
Overall, the study is a valuable contribution to the field and provides useful information for researchers interested in the application potential of oat. The authors should consider incorporating more specific examples and details to further enhance the clarity and usefulness of the review.
Response 3:
Thank you for your kind suggestion for improvement. According to your comment, we have provided more information and described in detail to further enhance the overall quality of our paper.

Reviewer 2 Report
The topic of the research article is of great interest. However, I would not recommend publishing the article in its current format as it requires lots of improvement. The main drawbacks of this manuscript
Below are several specific comments.
1. The English writing should be further improved, as there are some grammatical or typing errors. It is suggested to ask a native speaker to polish it.
2. The authors used around 84 references as it's well-known that a review paper should use the last 5 years of literature, to obtain the most relevant information. The percentages of the last 5 years of references were 40.48 % (34 references) while old references were 59.52 % (50 references)
3. The references should be extensive update
Author Response
Point 1:
The topic of the research article is of great interest. However, I would not recommend publishing the article in its current format as it requires lots of improvement. The main drawbacks of this manuscript
Below are several specific comments.
- The English writing should be further improved, as there are some grammatical or typing errors. It is suggested to ask a native speaker to polish it.
Response 1:
Thanks very much for your kind suggestion for language improvement of our paper. The English writing has been carefully checked and some mistakes have been revised. And for further improvement, the manuscript has been polished by a native-language labmate.
Point 2:
- The authors used around 84 references as it's well-known that a review paper should use the last 5 years of literature, to obtain the most relevant information. The percentages of the last 5 years of references were 40.48 % (34 references) while old references were 59.52 % (50 references)
Response 2:
We appreciate your professional review work on our manuscript. According to your suggestion, more literatures (totally 60 references) related which were published in the last 5 years have been involved. And the final percentages of the last 5 years of references are 63.16%.
Point 3:
- The references should be extensive update
Response 3:
Thanks a lot for your comment. The references’ part has been updated based on your suggestion.

Reviewer 3 Report
It is a review and it is about the fermented oat beverages, referring to the microorganisms and fermented techniques to obtain the fermented beverages. As a review it mentions several results about other authors that have worked with the fermented oat beverages of similar grains that are used to obtain fermented beverages. And the conclusions are consistent are well related to the information presented in the manuscript and other authors.
However, there is a figure #1 that provide a lot of information and the way it is presented is a little confused. What further controls should be considered? Use simple figures.
Author Response
Point 1:
It is a review and it is about the fermented oat beverages, referring to the microorganisms and fermented techniques to obtain the fermented beverages. As a review it mentions several results about other authors that have worked with the fermented oat beverages of similar grains that are used to obtain fermented beverages. And the conclusions are consistent are well related to the information presented in the manuscript and other authors.
However, there is a figure #1 that provide a lot of information and the way it is presented is a little confused. What further controls should be considered? Use simple figures.
Response 1:
Thanks a lot for your kind suggestion and we are sorry for our negligence to make such confusion. Figure 1 has been revised to be simple and more clearly presented according to your suggestion.

Reviewer 4 Report
Overall, the paper is interesting and valuable review on the fermentation of oat-based beverages. It adds on the previous review on the topic by Angel Angelov (https://doi.org/10.1007%2Fs13197-018-3186-y) especially by covering functional properties more widely.
However, there are severe problems with the review in terms of consistency of the language and the clear presentation of the subject. I propose a major revision of the paper to improve it. There are also some misunderstandings of the literature as indicated below.
Some minor notes:
Page 1 line 42: The mentioned 3 g beta-glucan per day is relevant only for cholesterol claim as indicated in the reference [7]. Current text implies that this dosage is relevant also for other health benefits
Page 2 line 60: “in antinutritional factors " It is not clear what is meant by this
Page 2 line 65: It is unclear what the “The current review” refers to. Obviously not to the current review paper made by the authors. Please revise.
Table 1 indicates that the data in the table is related to fermentation of oat products. In fact, many of the reference are not related to oats, but are related to other raw materials such as whey. Table should be revised so that fermentation studies related oats are emphasized.
Chapter 2.1 Lactic acid bacteria: this chapter should be rewritten and divided into separate paragraphs. Current version is very difficult to read. Furthermore, the chapter is very mechanical listing of studies and efforts should be made to make chapter more analytical.
Chapter 2.2 First sentence is very unclear, and it should be revised. Also, text in lines 188-191 should be revised
Chapter 3 lines 217-220: Text is very unclear, and it is not clear what is meant to refer to previous literature and what to the current study.
Lines 224-229 and 240: it is not clear if authors are talking about the endogenous enzymes in oats or exogenous enzymes added to the process.
Line 232: Should be “starch granules”
Table 2: It is not clear what is meant by de-branning.
It should be made clear, what type of the raw material in fermentation studies has been used (heat treated or enzyme active oats, flour, flakes etc)
Figure 3 in the current form is useless and does not demonstrate that the oligos remain unchanged during the fermentation and it is not possible to understand what is the message in the PCA plot.
Page 11 Line 357 what is meant with the term salt-degrading-enzymes? Do you mean bile-salts?
Author Response
Point 1:
Overall, the paper is interesting and valuable review on the fermentation of oat-based beverages. It adds on the previous review on the topic by Angel Angelov (https://doi.org/10.1007%2Fs13197-018-3186-y) especially by covering functional properties more widely.
However, there are severe problems with the review in terms of consistency of the language and the clear presentation of the subject.
Response 1:
Thanks a lot for your respective comment for English-writing improvement of our paper and we are very sorry for troubling you. The English writing of the manuscript has been carefully checked and some mistakes have been revised. And for further improvement and consistency of the language and clear presentation, the manuscript has been polished by a native-language labmate.
Point 2:
I propose a major revision of the paper to improve it. There are also some misunderstandings of the literature as indicated below.
Some minor notes:
Page 1 line 42: The mentioned 3 g beta-glucan per day is relevant only for cholesterol claim as indicated in the reference [7]. Current text implies that this dosage is relevant also for other health benefits
Response 2:
We sincerely appreciate your valuable suggestion. We have claimed in the original sentence that the dosage is only related to the production of cholesterol (line 41).
Point 3:
Page 2 line 60: “in antinutritional factors " It is not clear what is meant by this
Response 3:
Thanks for your comment. We have corrected this sentence as "Iron, zinc, magnesium, calcium or protein in oats can combine with phytic acid to form anti-nutritional factor, which is not conducive to human absorption. " based on the points you pointed out.
Point 4:
Page 2 line 65: It is unclear what the “The current review” refers to. Obviously not to the current review paper made by the authors. Please revise.
Response 4:
Thanks a lot for your comment. We have revised this sentence as “The previous researches on oats were mainly about oat raw materials, oat processing, etc., but rarely involved fermented oat beverages. "
Point 5:
Table 1 indicates that the data in the table is related to fermentation of oat products. In fact, many of the reference are not related to oats, but are related to other raw materials such as whey. Table should be revised so that fermentation studies related oats are emphasized.
Response 5:
Thank you for your advice to help make this manuscript even better. We have reorganized Table 1 to highlight the studies related to fermented oats.
Point 6:
Chapter 2.1 Lactic acid bacteria: this chapter should be rewritten and divided into separate paragraphs. Current version is very difficult to read. Furthermore, the chapter is very mechanical listing of studies and efforts should be made to make chapter more analytical.
Response 6:
Thanks for your suggestion. We have improved the chapter for more clarity of the intent and the readability of the text (lines 106-182).
Point 7:
Chapter 2.2 First sentence is very unclear, and it should be revised. Also, text in lines 188-191 should be revised
Response 7:
Thanks a lot for your kind suggestion. We have corrected this part as “Yeasts, mainly used in baking and brewing (wine, beer, bread fermentation) dominate the microbial composition of many alcoholic food fermentations” and “However, there are few studies on the application of yeasts in fermented oat beverages. It's possible that yeasts are not as capable of producing acid as LAB, or that the primary byproduct of yeasts consuming sugar is alcohol.” (lines 183-184,201-203).
Point 8:
Chapter 3 lines 217-220: Text is very unclear, and it is not clear what is meant to refer to previous literature and what to the current study.
Response 8:
I understand what you pointed out. We have modified this section as “Until now, the process technology researches of fermented oat beverages mainly concentrated on pre-treatment methods and fermentation conditions. Based on the available studies, the pre-treatment methods of fermented oat beverages (as shown in Figure 2) and the conditions in the fermentation process have been summarized.” (lines 232-236).
Point 9:
Lines 224-229 and 240: it is not clear if authors are talking about the endogenous enzymes in oats or exogenous enzymes added to the process.
Response 9:
Thanks a lot for your kind suggestion. The enzymes we are talking about is the exogenous enzymes added to the process, like amylase, protease, cellulases etc. In addition, we have modified the original text to avoid ambiguity. (lines 245,250).
Point 10:
Line 232: Should be “starch granules”
Response 10:
Thanks for pointing this out. We have revised it in the manuscript based on your comment.
Point 11:
Table 2: It is not clear what is meant by de-branning.
Response 11:
Thank you for your comment. De-branning, also known as pearling, is the process of sequentially removing the grain layers by the combined action of friction and abrasion.
Point 12:
It should be made clear, what type of the raw material in fermentation studies has been used (heat treated or enzyme active oats, flour, flakes etc)
Response 12:
We think this is an excellent suggestion. We have modified Table 1 by adding a supplement for the different type of the raw materials.
Point 13:
Figure 3 in the current form is useless and does not demonstrate that the oligos remain unchanged during the fermentation and it is not possible to understand what is the message in the PCA plot.
Response 13:
Thank you for your valuable comment. Figure 3 has been delated according to your suggestion.
Point 14:
Page 11 Line 357 what is meant with the term salt-degrading-enzymes? Do you mean bile-salts?
Response 14:
Thanks a lot for your suggestion. It does not mean bile-salts. We apologize for my misrepresentation. In the manuscript, we have revised the term as “a phytate-degrading enzyme”. The source of the term “a phytate-degrading enzyme” is due to phytase is a general term for a class of enzymes that catalyze the hydrolysis of phytic acid and its salts to inositol and phosphate (salts).

Round 2
Reviewer 2 Report
The authors have taken into consideration the most comments/suggestions of the reviewers during the revision of the manuscript.